# Mining host candidate regulators of schistosomiasis-induced liver fibrosis in response to artesunate therapy through transcriptomics approach

**Yajie Yuan[1,2⦿], Xinyue Lv[1⦿], Yahan Wu[1], Youhong Weng[1], Fangwei Dai[3], Haojie Ding[4], Riping Chen[5], Bin Zheng[4], Wenxia Zhao[5], Qunbo Tong[4], Jianzu Ding[4], Di Lou[4], Yunru Lai[6], Xiaofeng Chu[3], Longyou Zhao[6]\*, Shaohong Lu[4]\*, Qingming Kong◯[1,4]\***

**1** Key Laboratory of Biomarkers and In Vitro Diagnosis Translation of Zhejiang province, School of Laboratory Medicine and Bioengineering, Hangzhou Medical College, Hangzhou, China, **2** Department of Pathogen Biology, School of Basic Medicine, Nanjing Medical University, Nanjing, China, **3** Zhejiang Key Laboratory of Experimental Animal and Safety Evaluation, Hangzhou Medical College, Zhejiang Academy of Medical Sciences, Hangzhou, China, **4** School of Basic Medicine and Forensics, Key Laboratory of Bio-tech Vaccine of Zhejiang Province, Engineering Research Center of Novel Vaccine of Zhejiang Province, Hangzhou Medical College, Hangzhou, China, **5** School of Public Health, Hangzhou Medical College, Zhejiang Academy of Medical Sciences, Hangzhou, China, **6** Department of Laboratory Medicine, Lishui Second People's Hospital Affiliated to Wenzhou Medical University, Lishui, China

⦿ These authors contributed equally to this work.
\* zly8897@126.com (LZ); llsshh2003@163.com (SL); qmkong_1025@163.com (QK)

**Data Availability Statement:** The Raw RNA sequencing data that support the findings of this study are openly available in Sequencing Read Archive of the National Center for Biotechnology

## Abstract

### Background

Artesunate (ART) has been reported to have an antifibrotic effect in various organs. The underlying mechanism has not been systematically elucidated. We aimed to clarify the effect of ART on liver fibrosis induced by *Schistosoma japonicum* (*S. japonicum*) in an experimentally infected rodent model and the potential underlying mechanisms.

### Methods

The effect of ART on hepatic stellate cells (HSCs) was assessed using CCK-8 and Annexin V-FITC/PI staining assays. The experimental model of liver fibrosis was established in the Mongolian gerbil model infected with *S. japonicum* cercariae and then treated with 20 mg/kg or 40 mg/kg ART. The hydroxyproline (Hyp) content, malondialdehyde (MDA) content, superoxide dismutase (SOD) and glutathione peroxidase (GPX) activities in liver tissue were measured and histopathological changes of liver tissues were observed. Whole-transcriptome RNA sequencing (RNA-seq) of the liver tissues was performed. Differentially expressed genes (DEGs) were identified using bioinformatic analysis and verified by quantitative PCR (qPCR) and western blot assay.

Information (NCBI) (Project number PRJNA846574).

**Funding:** This study was supported by the National Natural Science Foundation of China (Grant No. 31501050 to QMK), Zhejiang Provincial Program for the Cultivation of High-Level Innovative Health Talents (Grant No. WJW2021002 to QMK), Basic Public Welfare Research Project of Zhejiang Province (Grant No. LGF22H180015 to FWD), Medical Health Science and Technology Project of Zhejiang Provincial Health Commission (Grant No. 2020KY530 to FWD),The S&T Major Project of Lishui (Grant No. 2023GYX03 to LYZ), Zhejiang Medical and Health Science and Technology Plan (Grant No. WKJ-ZJ-2203 to SHL), and the Central Leading Local Science and Technology Development Fund Project (Grant No. 2023ZY1019 to SHL). The funders had no role in study design, data collection and analysis, decision to publish, or preparation of the manuscript.

**Competing interests:** The authors have declared that no competing interests exist.

## Results

ART significantly inhibited the proliferation and induce the apoptosis of HSCs in a dose-dependent manner. *In vivo*, Hyp content decreased significantly in the ART-H group compared to the model (MOD) group and GPX activity was significantly higher in the ART-H group than in the MOD group. Besides, ART treatment significantly reduced collagen production (p <0.05). A total of 158 DEGs and 44 differentially expressed miRNAs related to ART-induced anti-schistosomiasis liver fibrosis were identified. The qPCR and western blot results of selected DEGs were consistent with the sequencing results. These DEGs were implicated in key pathways such as immune and inflammatory response, integrin-mediated signaling and toll-like receptor signaling pathways.

## Conclusion

ART is effective against liver fibrosis using Mongolian gerbil model induced by *S. japonicum* infection. We identified host candidate regulators of schistosomiasis-induced liver fibrosis in response to ART through transcriptomics approach.

## Author summary

Artesunate (ART) has been reported to have an antifibrotic effect while the underlying mechanism has not been systematically elucidated. Here, we evaluated the effect of ART on anti-hepatic fibrosis in vitro and in vivo. We found that ART significantly inhibited the proliferation and induced the apoptosis of HSCs. And also ART is effective against liver fibrosis using Mongolian gerbil model induced by *Schistosoma japonicum* (*S. japonicum*) infection. Hyp content decreased significantly in the ART-H group compared to the MOD group and GPX activity was significantly higher in the ART-H group than in the MOD group. Besides, ART treatment significantly reduced collagen production (p<0.05). A total of 158 DEGs and 44 differentially expressed miRNAs related to ART-induced anti-schistosomiasis liver fibrosis were identified through transcriptomics approach. These DEGs were implicated in key pathways such as immune and inflammatory response, integrin-mediated signaling and toll-like receptor signaling pathways. Our findings suggest that ART is effective against liver fibrosis. Furthermore, we identified 30 host candidate regulators in response to ART, and provides a basis for a better understanding of the molecular mechanisms of ART on anti-hepatic fibrosis.

## Introduction

Liver fibrosis is a very complex dynamic pathological process characterized by the activation of hepatic stellate cells (HSCs), immunity and inflammation, increased extracellular matrix (ECM) proteins and fibrogenesis, which involves the activation or inhibition of multiple signaling pathways in various cell types [1]. Major risk factors for liver fibrosis include hepatitis virus infection, *Schistosoma japonicum* infection, alcoholic liver disease, drugs and metabolic disorders [2]. Numerous studies on the mechanism of liver fibrosis have focused mainly on HSCs activation, and thus activated HSCs have been considered to be an antifibrotic target [3]. Although several drugs or small molecules are in clinical trials, very few have been approved for the treatment of liver fibrosis [4].

Dozens of natural compounds exert antifibrotic effects in various liver fibrosis models *in vivo* and *in vitro*. Fuzheng Huayu (FZHY), a traditional Chinese medicine approved by Chinese State Food and Drug Administration (NO: Z20050546) for treatment of liver fibrosis in China, consists of Salvia miltiorrhiza Bge., Cordyceps sinensis (Berk.) Sae., Prunus persica (L.) Batsch, Pinus massoniana Lamb., pentaphyllum (Thunb.) Makino and Schisandra chinensis (Turcz) Baill. FZHYcan attenuate liver fibrosis by inhibiting HSCs activation and inflammatory response, regulating ECM deposition and inhibiting hepatic sinusoidal capillarization and angiogenesis [5,6]. Recently, Luo et al showed that FZHY and artesunate (ART) could alleviate schistosomiasis-induced liver fibrosis by modulating mitochondrial autophagy [7]. ART, a stable derivative of artemisinin extracted from the stems and leaves of *Artemisia annua*, has multiple pharmacological effects, including anti-inflammatory, antitumor and antifibrotic activities [8–10]. Additionally, ART can inhibit fibrosis through different mechanisms involving inflammatory infiltration, autophagy of ferritin, collagen synthesis, HSCs apoptosis, etc. [11–13]. Several studies have also proven that ART prevents schistosomiasis-induced liver fibrosis by regulating mitochondrial function [7,14]. At present, existing studies on antifibrotic mechanisms of ART remain fragmented.

Mongolian gerbils are relatively suitable hosts of *S. japonicum*. The schistosomiasis-induced hepatic fibrosis in Mongolian gerbil is a very slow process because of continuous secreting soluble egg antigen (SEA) stimulation from eggs, which is similar to the process of pathological injury of the human liver [15]. Additionally, unlike the New Zealand rabbit model, the Mongolian gerbil model does not experience slow self-healing in the stage of advanced liver fibrosis [16]. It is the most realistic pathological reproduction of patients with infectious liver fibrosis.

Herein, we evaluated the effect of ART on liver fibrosis *in vitro* and *in vivo*. Subsequently, differentially expressed genes (DEGs) related to ART treatment were screened using whole transcriptome sequencing (RNA-seq) of liver tissues to explore potential regulatory mechanisms underlying the effect of ART on schistosomiasis-induced liver fibrosis.

## Methods

### Ethical approval of the study protocol

Animal experiments were approved by the Laboratory Animal Welfare and Ethics Committee of Zhejiang Laboratory Animal Center (Zhejiang, China).

### HSCs proliferation and apoptosis detection

The LX-2 cell line generated by Xu et al. and featured by the expression of intermediate filament proteins a-SMA, the marker for activated HSCs, as determined by immunocytochemistry and western analysis [17]. Activated HSCs are the main source of the ECM, and proliferation inhibition or apoptosis of activated HSCs can reduce ECM production and thus alleviate fibrosis. The Cell Counting Kit-8 (CCK-8) assay was used to detect the proliferation of ART-treated HSCs. Briefly, HSCs (2000 cells/well) were seeded into 96-well plates and exposed to different concentrations of ART (10, 20, 40, 60, 80, 100 and 120 μM), and the control and blank groups were treated with an equal volume of phosphate-buffered saline. After incubation for 24 h, each well was supplemented with 10 μL of CCK-8 solution and incubated for 1 h. Cell viability was determined by measuring the absorbance at 450 nm. Each experiment was performed in triplicate.

Apoptosis of HSCs was assessed by fluorescein isothiocyanate (FITC)/propidium iodide (PI) staining. Briefly, HSCs ($1 \times 10^4$ cells/well) were seeded into 24-well plates and exposed to different concentrations of ART (40 and 80 μM) for 48 h. Each well was supplemented with a mixture of 195 μL of Annexin V-FITC binding solution and 5 μL of Annexin V-FITC,

followed by the addition of 10 μL of PI pigment. After incubation for 10–20 min in the dark at 20–25˚C, the cells were observed under a fluorescence microscope.

## Animal model and ART treatment

Forty clean-grade male Mongolian gerbils (weighing 80 g) were used in the experiment. *S. japonicum* cercariae were induced and collected by exposing infected water-immersed snails to light for 1.5 h and counted by direct observation under a light microscope. To establish the animal model of schistosomiasis, 30 Mongolian gerbils were infected with 80 *S. japonicum* cercariae by the abdominal patch method. Praziquantel (PZQ), employed for decades as the key clinical agent to treat schistosomiasis, especially for adult worms, was used to eliminate the causative agent [18,19]. Eight weeks post-infection, Mongolian gerbils infected with cercariae were administered PZQ diluted in 1% sodium carboxymethyl cellulose (CMC-Na) solution at a dose of 75 mg/kg·d$^{-1}$ for 3 days via oral gavage. Twelve weeks post-infection, infected gerbils were randomly divided into three groups: model infection group (1% CMC-Na, MOD), low-dose ART treatment group (20 mg/kg·d$^{-1}$, ART-L) and high-dose ART treatment group (40 mg/kg·d$^{-1}$, ART-H). Mongolian gerbils in ART-L and ART-H groups were treated with 0.2 mL ART solution with a corresponding concentration once daily for 12 weeks via oral gavage. Mongolian gerbils in uninfected control (CON) and MOD groups were administrated 0.2 mL 1% CMC-Na solution once daily for 12 weeks via oral gavage.

## Measurement of biochemical indicators and Histological assessment

About 1 g of liver tissue was ground with 9 times of tissue homogenate and then centrifuged at 3,500 rpm/min for 10 min. Then, 10% saline homogenization suspension was used to detect glutathione peroxidase (GPX), superoxide dismutase (SOD) and malondialdehyde (MDA) (Nanjing Jiancheng Bioengineering Institute, Nanjing, China) and hydroxyproline (Hyp) (Beijing Solarbio Science & Technology Co., Ltd, Beijing, China) using standard commercial kits according to the manufacturer's instructions.

Tissue biopsy is used to assess the state of inflammation and fibrosis prior to ART administration. Briefly, the isoflurane-anesthetized Mongolian gerbils were fixed. After skin disinfection, approximately a 0.5 cm incision of skin was made using a sterilized scalpel. A small piece of liver tissue was removed from the upper part of the left lobe of the liver. The liver was covered with hemostat gauze containing heparin sodium to stop bleeding, and then the sutured wound surface was disinfected with iodophor. Light irradiation was used to restore body temperature. The gerbils were fasted for 12 h before and after surgery.

The superior segment of the left lateral lobe of each Mongolian gerbil liver was fixed in 10% formalin, embedded in paraffin, sectioned and stained with hematoxylin-eosin (H&E) and Masson's trichrome for light microscopy. The Ishak scoring system was utilized to evaluate the degree of liver inflammation and fibrosis [20]. Briefly, fibrosis was gauged as: 0 (no fibrosis), 1 to 2 (mild fibrosis), 3 to 4 (moderate fibrosis), 5 (sever fibrosis) and 6 (cirrhosis).

## Whole transcriptome sequencing

Total RNA was isolated from liver tissue (n = 2) by Trizol regent (Invitrogen, CA, USA). Then, RNA concentration and purity were measured using Nanodrop 2000. RNA integrity was assessed using agarose gel electrophoresis and Bioanalyzer 2100 (Agilent, CA, USA), and RNA integrity number (RIN) >7.0 was considered sufficient for microarray analysis. A small RNA library was constructed using TruSeq Small RNA Sample Prep Kits (Illumina, San Diego, USA), and single-end sequencing was performed on Illumina Hi-seq 2500. For strand-specific library construction and paired-end sequencing on the Illumina Hi-seq 4000, RNA samples

were purified using the Epicentre Ribo-Zero Gold Kit (Illumina, San Diego, USA) and fragmented into small pieces. The cleaved RNA fragments were then reverse-transcribed to create the final complementary DNA (cDNA) library using the RNA-seq sample preparation kit (Illumina, San Diego, USA) according to the manufacturer's protocol.

## Data processing and Bioinformatics analysis

Raw reads were processed by Cutadapt to generate clean reads and verified through FastQC (http://www.bioinformatics.babraham.ac.uk/projects/fastqc/). Clean reads were then mapped to the reference genome of the Mongolian gerbil (https://www.ncbi.nlm.nih.gov/genome/? term=Mongolian%20gerbil) and assembled using StringTie. Transcriptomes from Mongolian gerbil samples were merged to reconstruct a comprehensive transcriptome using Perl scripts. The transcript-level expression of RNA-seq was analyzed by calculating fragments per kilobase of exon per million mapped fragments (FPKM). DEGs with log2 fold change (FC) >1 or log2 FC < -1 and p< 0.05 were selected by the R package Ballgown. Gene ontology (GO) terms and the Kyoto encyclopedia of genes and genomes (KEGG) pathways of these DEGs were annotated. The protein-protein interaction (PPI) network of DEGs was performed using the STRING online analysis tool (https://string-db.org/). The PPI network was visualized and analyzed by Cytoscape software 3.7.2, and the top 30 DEGs were selected as hub genes. For microRNA (miRNA) analysis, unique sequences with 18–26 nucleotides were mapped to specific species precursors in miRBase 21.0 using the BLAST search to identify known miRNAs and novel 3p- and 5p-derived miRNAs. The prediction of miRNA target genes was carried out by OmicStudio tool at https://www.omicstudio.cn/tool based on TargetScan (5.0) and Miranda (3.3a), and the screening threshold was TargetScan_score ≥50 and miranda_Energy < -10.

## Quantitative PCR (qPCR) and Western blot assay

To validate RNA-seq results, total RNA was extracted from liver tissue using Trizol regent. Reverse transcription for each sample was performed on 200 ng of total RNA using a cDNA reverse transcription kit (Takara, Beijing, China). Thereafter, real-time fluorescent qPCR was utilized to detect the expression levels of the candidate genes. The primers were designed with Primer 5.0 software (S1 Table) and synthesized by Tsingke Biotechnology Co., Ltd (Beijing, China). The relative gene expression was calculated using the $2^{-\Delta\Delta Ct}$ method and GAPDH was used as an internal reference gene for each example.

The total protein of the liver tissue was extracted using a total protein extraction kit. Total protein concentration was determined using the BCA kit (Sheng gong, Shanghai, China). Then, 30 μg of each protein sample was denatured at 100˚C for 10 min and separated by sodium dodecyl-sulfate polyacrylamide gel electrophoresis (SDS-PAGE). Protein samples were then transferred to polyvinylidene difluoride membranes, blocked with 5% non-fatty milk for 1 h and blotted with relative antibodies (Proteintech, Wuhan, China). The enhanced chemiluminescence reagent was used for color development. The Alpha software processing system was used to analyze the optical density values of the target band.

## Statistics analysis

Data were presented as the mean ± SEM. Statistical analyses were performed using SPSS 26.0 and Graph Pad Prism 8.4 software. Statistical comparison between groups was performed using one-way analysis of variance (ANOVA) with a Tukey test and student's t-test. p< 0.05 was considered statistically significant.

## Results

### Effect of ART on HSCs and the animal model

The effect of ART on HSCs was investigated using CCK-8 and FITC/PI double staining assays. After ART treatment, apoptotic morphological changes were observed under the microscope. Results showed that ART (10 μM) significantly inhibited HSCs proliferation at 24 h after treatment (p< 0.05 vs. control) (**Fig 1A and 1B**). Meanwhile, the number of apoptotic cells gradually increased with the increase in ART concentration (**Fig 1C**). **Fig 2A** shows the respective timeline of the *in vivo* studies. After 12 weeks of ART treatment, Hyp content decreased significantly (p< 0.01) and GPX activity increased significantly in the ART-H group compared to the MOD group. While changes in MDA content and SOD activity did not show a statistical significance (**Fig 2B**). The livers in the MOD group were chestnut brown and had slightly increased volume and blunt edges. There was a marked improvement in liver morphology and color in the ART-H group; however, changes to the liver index were not significant (**S2 Table**). Histopathological changes in livers were assessed by H&E and Masson's trichrome staining. Compared with the CON group, inflammatory cell infiltration and collagen fiber deposition increased significantly in the MOD group (p< 0.05), while collagen deposition decreased significantly in the ART-H group, indicating that 40 μM of ART has a significant protective effect against liver fibrosis (p< 0.01). In addition, we compared the scores before and after ART treatment within the group, and found that after several weeks, the changes of inflammation and fibrosis in the MOD group and the ART-L group were not obvious, and the fibrosis level of the ART-H group decreased compared with before administration (**Fig 2C**).

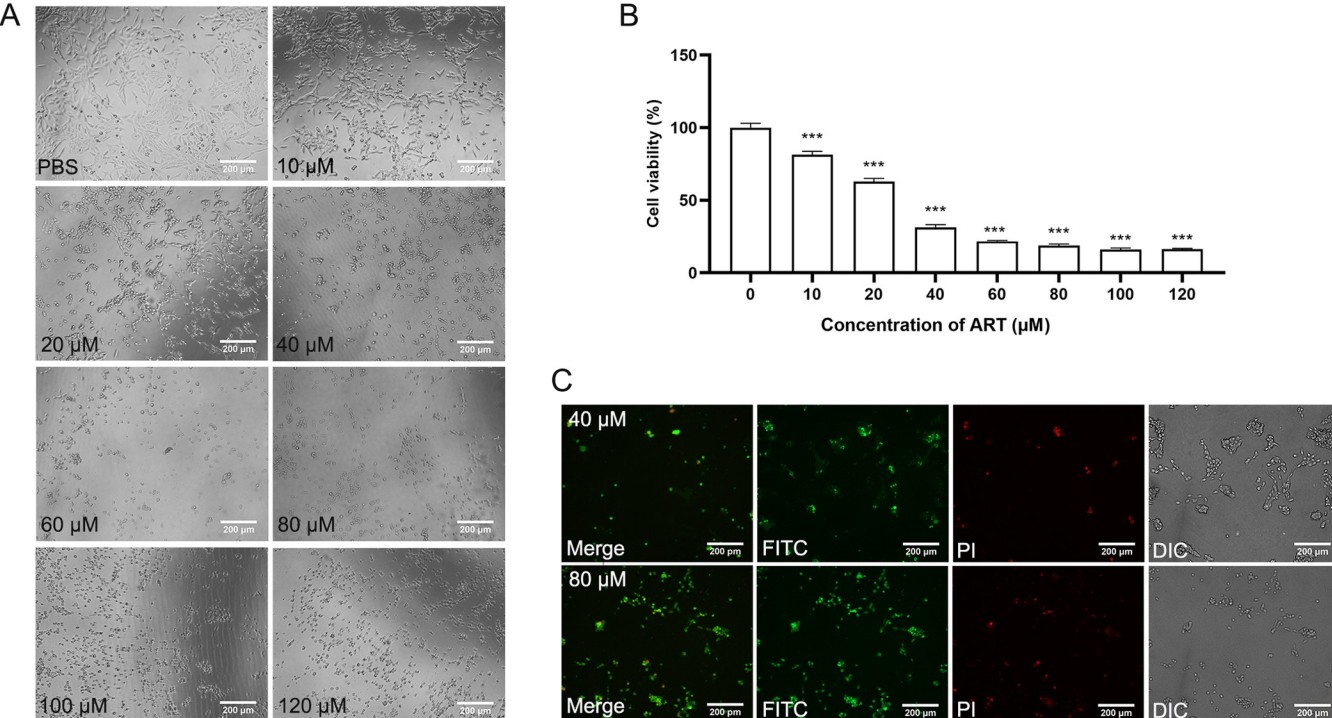

**Fig 1. Effect of ART on HSCs.** (A) HSCs morphology after 24 h of treatment with different ART concentrations under an inverted phase-contrast microscope (100× magnification). (B) Viability of HSCs calculated at 450 nm absorbance value with different concentrations of ART for 24 h. (C) Fluorescence microscopy images of apoptotic cells after 24 h of treatment with different concentrations of ART, at 100× magnification. Data are presented as mean ± SEM (n = 3). ***p< 0.001 vs CON group.

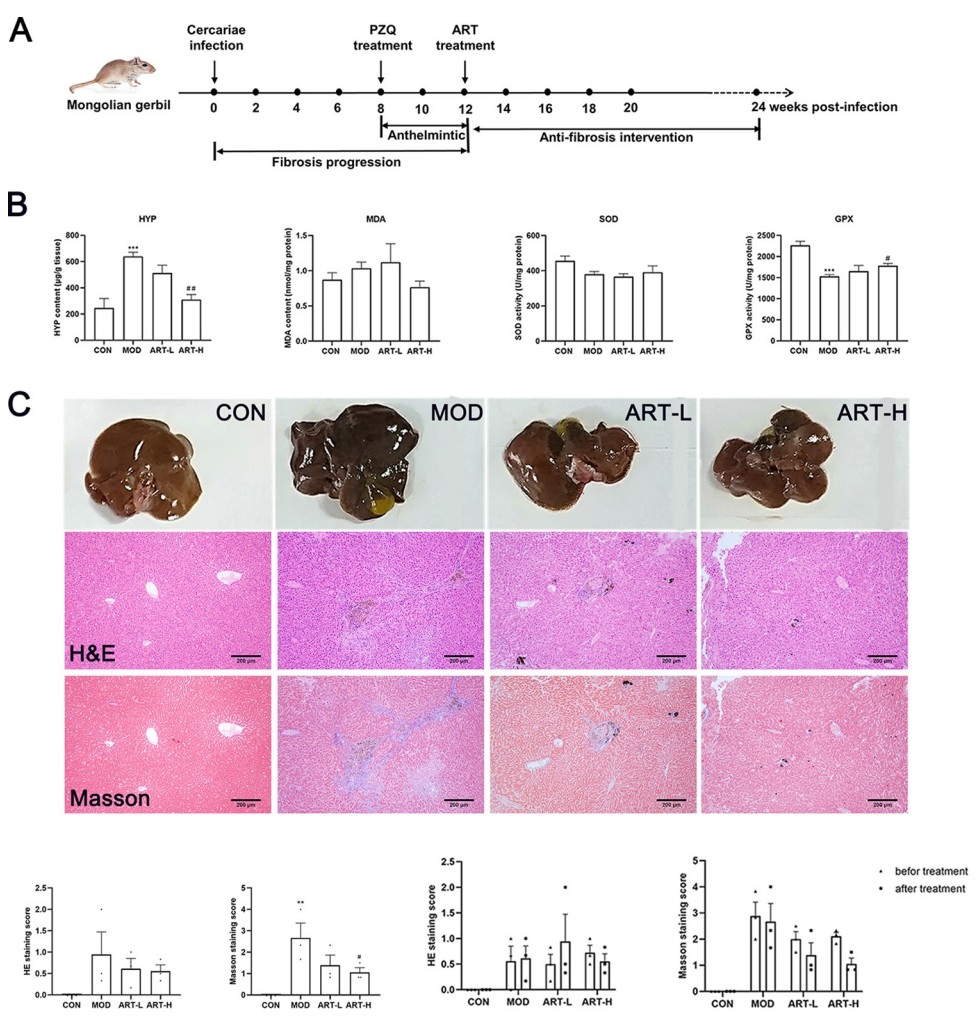

**Fig 2. Effect of ART on schistosomiasis-induced liver fibrosis in the Mongolian gerbil.** (A) Timeline of *in vivo* studies. (B) Levels of Hyp, MDA, SOD, and GPX. (C) Liver morphological and pathological changes of liver tissues (100× magnification), Statistical analysis of HE and Masson staining scores after ART treatment (left). Statistical analysis of HE and Masson staining scores before and after ART treatment (right). Data are presented as mean ± SEM (n = 3). ** $p < 0.01$ and *** $p < 0.001$ vs CON group. # $p < 0.05$ vs MOD group).

## Transcriptome expression profiles of schistosomiasis-induced liver fibrosis in response to ART

For each sequencing library, the reads of the Q20, Q30 and GC content exceeded 99%, 96% and 47% respectively, which indicated a high quality of clean data (**S3 Table**). Meanwhile, over 90% of the total reads were successfully mapped to the reference genome (**S4 Table**). Successfully mapped reads were then used for subsequent analysis. The DEGs were selected as log2 (FC) > 1 or log2 (FC)<-1 and with a P value < 0.05 in the Ballgown R package. Then DEGs were screened for subsequent analysis. Compared with the CON group, 467 genes were up-regulated and 144 genes were down-regulated in the MOD group, while 187 genes were up-regulated and 142 genes were down-regulated in the ART group. Compared ART group with the MOD group, 117 genes were up-regulated and 216 genes were down-regulated in the ART group (**Fig 3A**). The Venn diagram showed the shared and unique DEGs between different groups. Except for ART vs CON, DEGs between MOD vs CON and ART vs MOD were

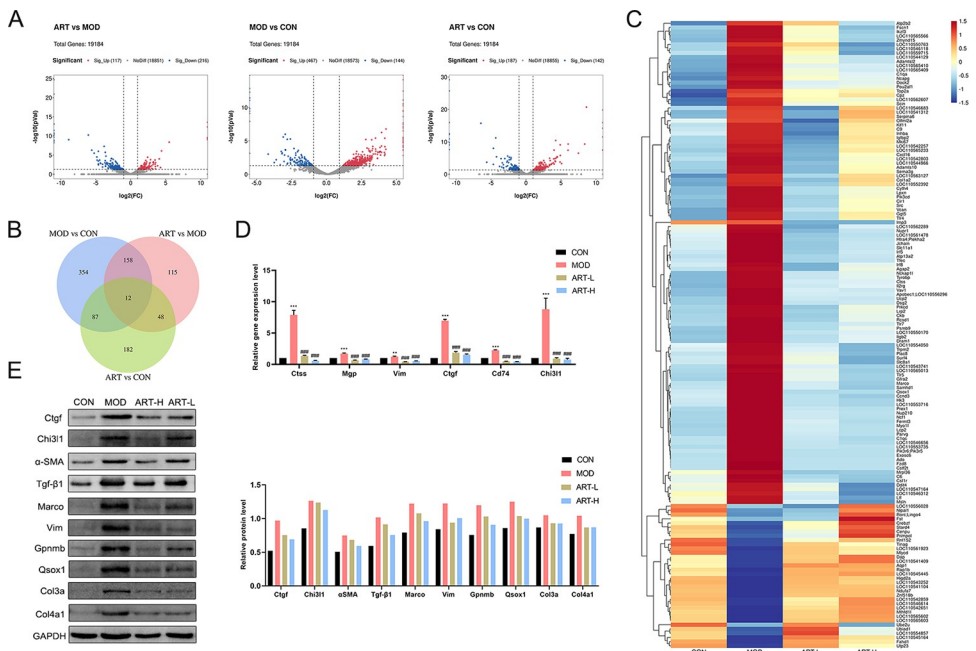

**Fig 3. Screening and validation of DEGs.** (A) Volcano plot, (B) Venn diagram and (C) Heat map of DEGs. (D) qPCR validation of DEGs. (E) Western blot validation of DEGs. Data are presented as mean ± SEM (n = 2). ** p< 0.01 and *** p< 0.001 vs CON group; ### p< 0.001 vs MOD group.

considered ART treatment-related genes, which revealed 158 DEGs related to the ART effect against schistosomiasis-induced liver fibrosis (**Fig 3B**). The heat map revealed differential genes expression patterns between different groups (**Fig 3C**). Meanwhile, a total of 44 differentially expressed miRNAs related to ART-induced anti-schistosomiasis liver fibrosis were identified (**S1 Fig**). We selected molecules closely related to fibrosis progression to validate sequencing data by qPCR and WB assays, and found that the expression of pro-fibrosis related molecules (such as Ctgf, α-SMA, Tgf-β1, Col3a and Col4a1) decreased in the ART-treated group compared with the untreated MOD group. The gene and protein expression levels of pro-fibrosis factors such as Ctgf, α-SMA, TGF-β1, Vim, and collagen detected by qPCR and western blot were consistent with sequencing results, indicating that sequencing results are reliable (**Fig 3D and 3E**).

## Enrichment analysis of DEGs and PPI network construction

Subsequently, GO and KEGG enrichment analyses of the 158 DEGs specified above to be ART-related effects were performed respectively to explore the potential mechanism of ART. The top 20 significant GO and KEGG pathway terms are displayed in **Fig 4A and 4B**. GO analysis showed that these DEGs were mainly enriched in immune response, inflammatory response and integrin-mediated signaling pathways. (**Fig 4A**). KEGG pathway analysis revealed that these DEGs were mainly enriched in complement and coagulation cascades, chemokine signaling pathway, FC gamma R-mediated phagocytosis, focal adhesion, platelet activation, toll-like receptor signaling pathway and HIF-1 signaling pathway (**Fig 4B**). The PPI network of the 158 DEGs generated 109 nodes and 271 edges (p< 1.0–16), and the top 30 DEGs were selected as hub genes (**Fig 4C**). Most of the predicted target genes of 6 miRNAs (miR-193a-3p, miR-370-3p, miR-365-3p, miR-2137, miR-3473b and miR-3473e) were also differentially expressed hub genes related to ART-induced anti-schistosomiasis liver fibrosis,

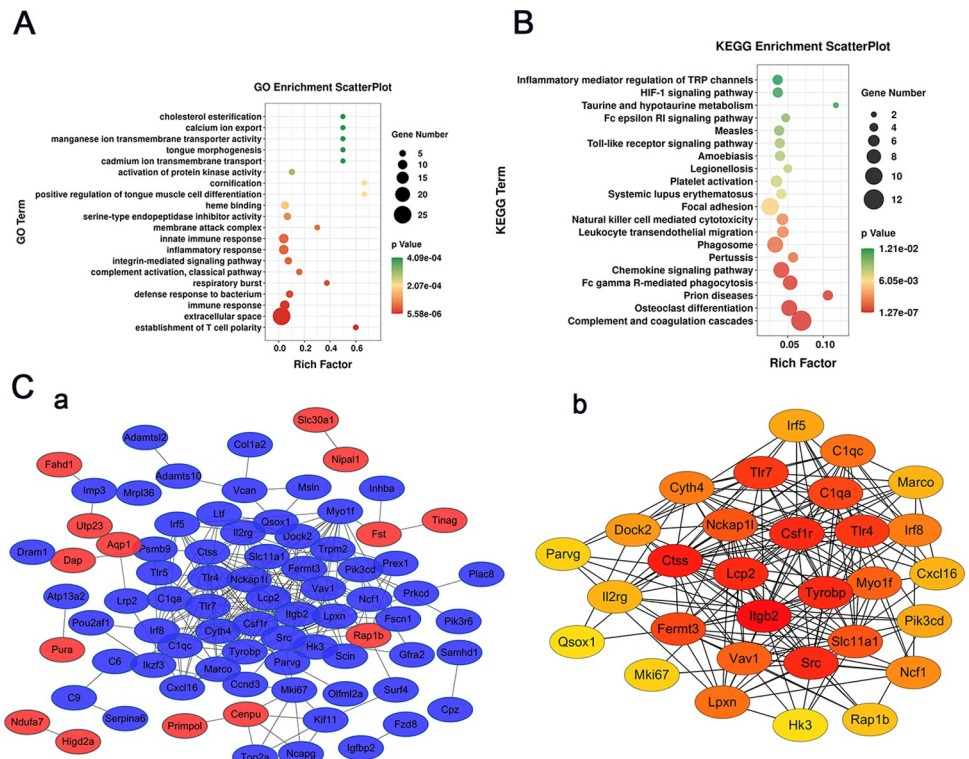

**Fig 4. Enrichment analysis of DEGs and PPI network construction.** (A) GO enrichment analysis. (B) KEGG pathway enrichment analysis. (C) a, PPI analysis; blue represents down-regulated genes, and red represents up-regulated genes. b, Top 30 hub genes based on the degree algorithm of cytoHubba; dark red represents a high degree of connectivity.

including Tlr4, Cyth4, Irf8, Nckap1l, Myo1F, Src, Fscn1, Adamtsl2, Prkcd, Surf4 and Ddit4 (**S5 Table**). These DEGs were implicated in key pathways, such as immune and inflammatory response, integrin-mediated signaling and toll-like receptor signaling pathways.

## Discussion

Schistosomiasis, including cases caused by *S*. japonicum, mainly leads to liver fibrosis and advanced diseases, such as portal hypertension [21]. PZQ has been approved for the treatment of schistosomiasis because of its high effectiveness. Oral administration of PZQ at a single dose of 400 mg/kg resulted in worm reduction rates of 88–100% in mice [22], which indicated that our dose (75 mg/kg for 3 days) can effectively kill worms. Additionally, ART can effectively kill juvenile Schistosoma and has a very good effect of preventing schistosomiasis, so it couldn't develop into adult worm and lays eggs [23]. However, previous studies demonstrated that the removal or elimination of the etiological agents may not be sufficient to stop liver fibrosis progression [24].

ART has received considerable attention because of its various pharmacological properties. Although the antifibrotic effect of ART has been proven, the specific molecular mechanism has not been systematically elucidated [25]. The present study investigated the effect of ART on liver fibrosis in HSCs and the *S. japonicum*-infected Mongolian gerbil model. Mechanistically, RNA-seq analysis of liver tissue was performed to explore the potential molecular mechanism of ART in liver fibrosis.

Our data showed that ART significantly inhibited the proliferation and induced the apoptosis of HSCs *in vitro*, and effectively alleviated schistosomiasis-induced liver fibrosis *in vivo*. In HSCs, ART significantly inhibited cell proliferation in a dose-dependent manner in a concentration range of 10–20 μM. In addition, the number of apoptotic cells gradually increased with the increase in the concentration of ART, indicating that ART promotes cell apoptosis in a dose-dependent manner. A recent study also showed that ART promotes HSCs apoptosis by inhibiting mitochondrial function [14]. Moreover, we successfully used a Mongolian gerbil model of liver fibrosis induced by *S. japonicum* cercariae infection to explore the antifibrotic activity of ART *in vivo*. A study on the acute toxicity of intravenous ART in rats showed that the LD50 value was 488 mg/kg in rats [26]. Our previous long-term toxicity study showed that the LD50 of artesunate given orally to rats once daily for 28 d was 150 mg/kg/d and the safe dose was 25 mg/kg/d [27]. Therefore, to gerbils, we chose a high dose of 40 mg/kg, which is relatively safe in animal experiments. It was found that gavage ATR treatment slightly improved the liver index and reduced inflammatory cell infiltration and collagen deposition in the ART-H group, which is consistent with findings of a previous study that reported a decreased area of fibrosis and granuloma in the ART intervention group in a mouse model [14]. There is a limitation in our study that live eggs were not tested after PZQ treatment and before ART treatment, which leads to the conclusion that the antifibrotic effect of ART is not due to its killing effect on eggs is not convincing. However, studies have shown that ART is effective against juvenile schistosomula, and has a very good effect of preventing schistosomiasis, so it will not cause the survival and expulsion of eggs [23]. Also, no live adult worms or eggs were found by tissue biopsy before ART administration. In addition, PZQ did not reduce fibrosis in the MOD group, which indicates that the reduction of fibrosis in the ART group was due to its antifibrotic effect rather than its antimalarial effect. Collectively, these results suggest that ART may relieve schistosomiasis-induced liver fibrosis to a certain extent.

To further explore the antifibrotic mechanism of ART, we performed transcriptome RNA-seq of the liver from CON, MOD and ART groups. Further, bioinformatics analysis identified 158 DEGs related to ART-induced anti-schistosomiasis liver fibrosis. GO function analysis showed that these DEGs were involved in immune response, extracellular space and calcium ion transportation terms. KEGG pathway analysis revealed that the DEGs were significantly enriched in immune response, focal adhesion, platelet activation and HIF-1 signaling pathways.

Current treatment strategies for liver fibrosis directly inhibit HSCs activation, induce activated HSCs apoptosis, increase ECM degradation and suppress inflammation (**Fig 5**). The activation of HSCs is crucial in liver fibrosis, and TGF-β1 is responsible for HSCs activation. The Src family of tyrosine kinases are activated in fibrosis of various organs such as the lung, kidney and liver, and are considered to be the amplifiers of TGF-β1 signal in the pathological process of fibrosis [28]. The activated Src under hypoxic conditions of tissue cells can mediate the expression of Ctgf induced by TGF-β1, which further stimulates collagen secretion, eventually accelerating the fibrotic process [29,30]. At present, Src has become a potential therapeutic target for various organ fibrosis. For example, Sarakatinib, a tyrosine kinase activity inhibitor, has been approved by the Food and Drug Administration for the treatment of idiopathic pulmonary fibrosis [31,32]. Our RNA-seq results showed that Src was significantly down-regulated in the ART group compared with the MOD group, which indicated that Src is a potential target of ART in liver fibrosis [33]. Additionally, previous studies reported that ART prevents schistosomiasis-induced liver fibrosis by regulating mitochondrial function in HSCs [7,14]. Six mitochondrial-related DEGs (Ddit4, Ndufa7, Higd2a, Ucp2, Mrpl36 and Fahd1) were identified in the present study, which are potential targets of ART. The Ddit4 is expressed under stress and is located downstream of HIF-1α, which negatively regulates the mammalian

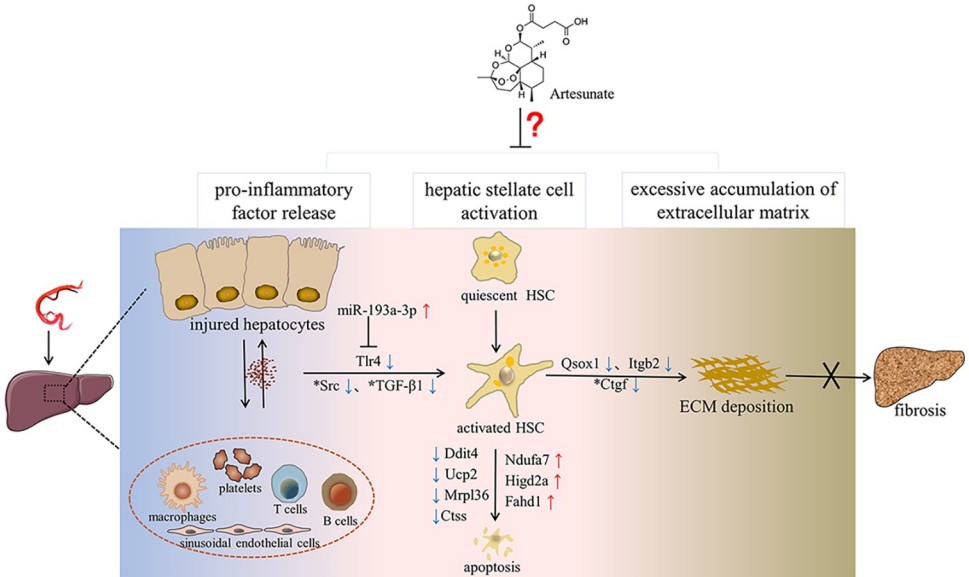

**Fig 5. Potential candidate regulators of schistosomiasis-induced liver fibrosis in response to ART therapy.** Red arrows indicate up-regulated genes and green arrows indicate down-regulated genes; an asterisk indicates previously reported genes. "X" indicates blockage of the fibrotic process. ART may alleviate liver fibrosis in the gerbil model by reducing inflammation, inhibiting HSCs activation and decreasing collagen deposition. Liver fibrosis induced by *Schistosoma japonicum* infection is the result of the deposition of eggs in the liver. Matured eggs cause blood vessel and hepatocyte damage by SEA, which triggers egg granuloma with a complex immune inflammatory reaction regulated by damaged hepatocytes, thereby recruiting immune cells. ART reduced liver inflammation possibly by up-regulating miR-193a-3p that may inhibit TLR4 signaling, and inhibiting producing profibrotic cytokines such as TGF-β1, thereby inhibiting HSCs activation. ART may also induce apoptosis of activated HSCs through mitochondrial pathways. It possibly decreases the ECM deposition by down-regulating genes associated with collagen synthesis such as Ctgf, Qsox1 and Itgb2.

target of rapamycin (mTOR)-induced cell proliferation and induces autophagy [34,35]. Ctss is one of the top 30 DEGs identified in this study. It promotes cell migration and invasion by regulating $Ca^{2+}$ homeostasis and ECM degradation [36]. Studies have shown that inhibiting Ctss may increase the expression of mitochondrial calcium uniporter and enhance the absorption capacity of mitochondrial $Ca^{2+}$, leading to mitochondrial $Ca^{2+}$ overload, the production of large amounts of ROS and mitochondria-mediated cell apoptosis [37]. However, further *in vivo* experimental verification is required to confirm this interesting hypothesis.

Excessive deposition of ECM is the main pathological feature of liver fibrosis, and thus increasing ECM degradation is a pivotal strategy for antifibrotic therapies. RNA-seq and PPI analysis showed that several DEGs, such as Ctgf, Qsox1, Fscn1, Myo1f, Lpxn, Scin, Parvg, Itgb2 and Fermt3, may participate in the regulation of actin cytoskeleton assembly and ECM adhesion [38–41]. Interestingly, Ctgf has been reported to be up-regulated in various fibrotic diseases, including lung, kidney, skin and liver fibrosis [42,43]. Inhibition of Ctgf inactivated HSCs and reduced ECM formation [44,45]. Therefore, Ctgf may be a potential therapeutic target for reversing liver fibrosis.

## Conclusion

In summary, we successfully used a Mongolian gerbil model of liver fibrosis induced by *S. japonicum* cercariae infection, mimicking human infection. The efficacy test showed that ART effectively prevented liver fibrosis *in vitro* and *in vivo*. ART may effectively alleviate schistosomiasis-induced liver fibrosis by reducing inflammation and collagen fiber deposition. We

established the first comprehensive transcription profile, mined host candidate regulators and provided an overview of the complex network of the mechanism of schistosomiasis-induced liver fibrosis in response to ART. The findings of this study are essential to understanding the mechanism of action of ART and verifying potential diagnostic markers or novel therapeutic agents for schistosomiasis-induced liver fibrosis.

## Supporting information

**S1 Fig. Differentially expressed miRNAs.** (A) Heat map of differentially expressed miRNAs and (B) Venn diagram of differentially expressed miRNAs.
(TIF)

**S1 Table. Primer information.**
(DOC)

**S2 Table. Liver mass index.**
(DOC)

**S3 Table. Quality parameters of clean data.**
(DOC)

**S4 Table. Number of successfully mapped reads.**
(DOC)

**S5 Table. MiRNAs target gene prediction results.**
(DOCX)

## Author Contributions

**Conceptualization:** Yajie Yuan, Qingming Kong.

**Data curation:** Yajie Yuan, Xinyue Lv.

**Funding acquisition:** Fangwei Dai, Haojie Ding, Longyou Zhao, Shaohong Lu, Qingming Kong.

**Investigation:** Qingming Kong.

**Methodology:** Wenxia Zhao, Qunbo Tong, Jianzu Ding, Di Lou, Xiaofeng Chu.

**Project administration:** Qingming Kong.

**Resources:** Fangwei Dai, Yunru Lai, Longyou Zhao.

**Supervision:** Longyou Zhao, Shaohong Lu, Qingming Kong.

**Validation:** Yahan Wu, Youhong Weng, Haojie Ding, Riping Chen, Bin Zheng.

**Visualization:** Yajie Yuan, Xinyue Lv.

**Writing – original draft:** Yajie Yuan.

**Writing – review & editing:** Yajie Yuan, Xinyue Lv, Qingming Kong.

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
