## [Decision Letter · Decision Letter 0]

31 Oct 2022

Dear Mr. Kong,

Thank you very much for submitting your manuscript "Mining Host Candidate Regulators of Schistosomiasis Liver Fibrosis in Response to Artesunate Therapy through Transcriptomics Approach" for consideration at PLOS Neglected Tropical Diseases. As with all papers reviewed by the journal, your manuscript was reviewed by members of the editorial board and by several independent reviewers. In light of the reviews (below this email), we would like to invite the resubmission of a significantly-revised version that takes into account the reviewers' comments. 

The reviewers commented that this was an interesting study, but that substantial editing of the manuscript is required before being suitable for publication. In particular specific attention is needed in the methods and results sections, providing more information as to the methodology used and expanding the figure legends as detailed below. Please address all the comments raised by the reviewers. One reviewer made substantial comments to the manuscript text in addition to that detailed below. If you have not received this document, please make sure to request it from the journal editing staff.

We cannot make any decision about publication until we have seen the revised manuscript and your response to the reviewers' comments. Your revised manuscript is also likely to be sent to reviewers for further evaluation.

Sincerely,

Krystyna Cwiklinski, PhD

Academic Editor

Cinzia Cantacessi

Section Editor

The reviewers commented that this was an interesting study, but that substantial editing of the manuscript is required before being suitable for publication. In particular specific attention is needed in the methods and results sections, providing more information as to the methodology used and expanding the figure legends as detailed below. Please address all the comments raised by the reviewers. One reviewer made substantial comments to the manuscript text in addition to that detailed below. If you have not received this document, please make sure to request it from the journal editing staff.

Reviewer's Responses to Questions

**Key Review Criteria Required for Acceptance?**

**Methods**

-Are the objectives of the study clearly articulated with a clear testable hypothesis stated?

-Is the study design appropriate to address the stated objectives?

-Is the population clearly described and appropriate for the hypothesis being tested?

-Is the sample size sufficient to ensure adequate power to address the hypothesis being tested?

-Were correct statistical analysis used to support conclusions?

-Are there concerns about ethical or regulatory requirements being met?

Reviewer #1: Please see the comments I have made on the manuscript, that should be carefully addressed. In several places, additional details regarding materials and methods have been requested. These questions pertain to the design of the study, amount of replication involved, and applicability of some of the bioinformatics approaches. I don't think I have ethical concerns, but would like the level of dosing with ART to be clarified (amount per day, for how many days), and an explantion offered if this is generally considered to be a safe dose, especially at the high level used. The overall effort was hampered by not defining ALL their acronyms clearly - this was a distraction for me.

Reviewer #2: See summary and General comments

Reviewer #3: The expression trend of qPCR targeted Ctss, Mgp, Vim, Ctgf, Cd74, Chi3l1 was consistent with the sequencing results. 

These genes need to be validated by WB.

**Results**

-Does the analysis presented match the analysis plan?

-Are the results clearly and completely presented?

-Are the figures (Tables, Images) of sufficient quality for clarity?

Reviewer #1: Again, several requests for additional details have been made on the manuscript. In particular, the figure legends are too brief and need more explanation. Some of the effects noted for ART seem to be marginally significant, even at what seems like high treatment doses. The results of the transcriptome study seem to return an interesting and relevant set of molecules that can plausibly be linked to the fibrosis phenomenon.

Reviewer #2: See summary and General comments

Reviewer #3: A scale bar or magnification should be indicated on the picture.

**Conclusions**

-Are the conclusions supported by the data presented?

-Are the limitations of analysis clearly described?

-Do the authors discuss how these data can be helpful to advance our understanding of the topic under study?

-Is public health relevance addressed?

Reviewer #1: The conclusions seem to be supported by the data, but some of the limitations noted above with respect to further experimental details will help to clarify their conclusions. Some limitations are mentioned by the authors and this makes sense as they have uncovered new candidate molecules inbvolved in fibrosis, but I have posed some questions along these lines on the manuscript that need to be addressed. There is a broad justification for wanting to better control or reduce fibrosis as it relates to diseases like schistosomiasis. To give their paper more clarity and to potentially increase its significance, it would be helpful for the authors to briefly indicate why fibrosis is bad, but might it be the case that when prevented, it might actually worsen a patient's prospects by facilitating schistosome egg survival and its secretion of harmful substances? It is understood though that exploration of ART and its potential role in controlling fibrosis could have useful applications. I would like the authors to discuss whether the doses required in their study to achieve an effect are practical for human application.

Reviewer #2: See summary and General comments

Reviewer #3: (No Response)

**Editorial and Data Presentation Modifications?**

Reviewer #1: Throughout the English is off enough to be a distraction in some key places. Some suggestions are found in my comments but I have not edited it thoroughly for English grammar. Thefigure legends need to be more thorough.

Reviewer #2: See summary and General comments

Reviewer #3: (No Response)

**Summary and General Comments**

Reviewer #1: I was intrigued by the fact that ART might have beneficial applications to schistosome-mediated fibrosis so am sympathetic to the overall effort. The presentation was weakened somewhat by the lack of details in methods and explanation of results. Clarity with respect to the doses required to achieve effects and how effectively transcriptomics data from gerbils can be interpreted using a mouse reference genome is particularly needed. Several comments were made directly on the manuscript itself that should be consulted.

Reviewer #2: Yuan an collaborators explored the potential of Artesunate, a well known antimalarial with anthelminthic potential, to ameliorate liver fibrosis as a result of S japonicum infection in Jirds and the liver tissue analyzed by transcriptomics to unveil possible mechanistic bases for such an effect.

ART were reported to inhibit proliferation and induce apoptosis of Hepatic stellate cells in a dose-dependent manner. Levels of HA, ALT and inflammatory cell infiltration were claimed to be decreased after ART administration in vivo with reduction of collagen fibers. Transcriptomic analyses of the liver revealed a set of differentially expressed genes and miRNAs where some hits could be validated for differential expression b qPCR. Interestingly, the authors report on a hub of the 30 most DEGs as mediators of major immune and inflammatory pathways. The authors went on to conclude on the ability of ART to prevent liver fibrosis in Jirds with a provision of candidate genes involved in the regulation of schistosomiasis liver fibrosis in response to ART.

The work is conceptually novel in its proceeding, specifically the use of transcriptomics to reveal candidate regulators of Artesunate-mediated regulation of S. japonicum driven liver fibrosis. The conclusions are appealing but would require considerable caution and additional control experimentation to conclusively support the present claims.

Major comments:

1. How certain are the authors that the PZQ treatment did completely clear the infection? Were worm / egg burden measured after PZQ treatment and at the end of the subsequent ART treatment to ensure that the reduction in the progression of liver fibrosis observed are as a result of the anti-fibrotic potential of ART and not mostly due to its additional antiparasitic potential on S japonicum ? (Xiao et al., Chin Med J (Engl). 1996 Apr;109(4):272-5; li et al, Acta Trop. Nov-Dec 2005;96(2-3):184-90.)

2. Figure 2B. HA and ALT levels should also be compared between the ART groups and the Model groups to reflect true reduction of these factors by ART treatment specifically. As it stands, HA and ALT levels do not appear to decay in ART groups when compared to Model groups arguing against a liver protecting effect of the ART treatment.

3. Figure 2C is highly qualitative, thus suggestive, and would benefit from a quantitative assay of measurement of tissue fibrosis such as hydroxyproline quantification. Moreover here, the staining scores do not reflect an overall reduction in the Model group when compared to the ART (after administration) group arguing strongly against an anti-fibrotic effect of ART in this study (critical! As this is the basis for the subsequent transcriptomic screening). One would have expected the ART groups after administration to be lower than the Model group even before administration. The continued increase of the fibrosis score in the Model group after administration when compared to the model group before administration does suggest an inability of the prior PZQ treatment to completely clear the schisto infection and would support that all effects observed with ART treatment might result from an anti-parasitic rather than anti-fibrotic action.

4. Why were markers of fibrosis not tested by qPCR as well ? i.e. a-SMA, Collagen etc… to support the anti-fibrotic claim of ART treatment in this particular study.

Minor comments;

1. Language is to be carefully checked throughout the manuscript for grammatical mistakes and typos eg. line 90 ‘…in the fluorescence microscope’ should read ‘…under the fluorescence microscope’; Line 121:’ Total RNA was isolated form…’ should read ‘ total RNA was isolated from…’

2. Caution should be made on exaggerated statements: ‘We developed liver fibrosis model in Mongolian gerbil mimicking human infection with S. japonicum cercariae’. This model is well known and merely reproduced by the authors. Eg. Liang et al., Int J Parasitol. 1983 Dec;13(6):531-8. doi: 10.1016/s0020-7519(83)80024-1.

3. How do the identified factors relate to previously reported factors in the literature using other omics approaches following ART treatment of S japonicum-infected rodents (Kong et al., Mol Biosyst. 2015 May;11(5):1400-9. doi: 10.1039/c5mb00074b.) Is a consensus observable here.

4. Harmonize the referencing style to that of Plos NTDs throughout the manuscript eg. Line 294 :’… that Src is a potential target of ART to liver fibrosis [23](Berkoz et al. 2021).’

Reviewer #3: (No Response)

PLOS authors have the option to publish the peer review history of their article (what does this mean?). If published, this will include your full peer review and any attached files.

Reviewer #1: No

Reviewer #2: Yes: Justin Komguep Nono

Reviewer #3: No
---

## [Decision Letter · Decision Letter 1]

20 Feb 2023

Dear Dr. Kong,

Thank you very much for submitting your manuscript "Mining Host Candidate Regulators of Schistosomiasis-Induced Liver Fibrosis in Response to Artesunate Therapy through Transcriptomics Approach" for consideration at PLOS Neglected Tropical Diseases. As with all papers reviewed by the journal, your manuscript was reviewed by members of the editorial board and by several independent reviewers. In light of the reviews (below this email), we would like to invite the resubmission of a significantly-revised version that takes into account the reviewers' comments. 

The reviewers still have concerns regarding this manuscript particularly relating to whether the claims the authors have made have been substantiated by their results/data. The authors need to address these comments before the manuscript can be considered for publication.

We cannot make any decision about publication until we have seen the revised manuscript and your response to the reviewers' comments. Your revised manuscript is also likely to be sent to reviewers for further evaluation.

Sincerely,

Krystyna Cwiklinski, PhD

Academic Editor

Cinzia Cantacessi

Section Editor

The reviewers still have concerns regarding this manuscript particularly relating to whether the claims the authors have made have been substantiated by their results/data. The authors need to address these comments before the manuscript can be considered for publication.

Reviewer's Responses to Questions

**Key Review Criteria Required for Acceptance?**

**Methods**

-Are the objectives of the study clearly articulated with a clear testable hypothesis stated?

-Is the study design appropriate to address the stated objectives?

-Is the population clearly described and appropriate for the hypothesis being tested?

-Is the sample size sufficient to ensure adequate power to address the hypothesis being tested?

-Were correct statistical analysis used to support conclusions?

-Are there concerns about ethical or regulatory requirements being met?

Reviewer #1: See my comments on the attached manuscript and general remarks below. The revised version I viewed did not alleviate concerns regarding design or sample sizes, but it did alleviate my concern about doses of ART used.

Reviewer #2: see general comments

Reviewer #3: (No Response)

**Results**

-Does the analysis presented match the analysis plan?

-Are the results clearly and completely presented?

-Are the figures (Tables, Images) of sufficient quality for clarity?

Reviewer #1: See comments on the manuscript. The results often lacked detail and the details of the methods used to get them in some cases were not provided. Gaps exist in the presentation. Figures were clear, but not fully explained by the legends.

Reviewer #2: see general comments

Reviewer #3: (No Response)

**Conclusions**

-Are the conclusions supported by the data presented?

-Are the limitations of analysis clearly described?

-Do the authors discuss how these data can be helpful to advance our understanding of the topic under study?

-Is public health relevance addressed?

Reviewer #1: Unfortunately, the main conclusions are not strongly supported. The authors do a good job in putting the results they have into a reasonable framework or model.

Reviewer #2: see general comments

Reviewer #3: (No Response)

**Editorial and Data Presentation Modifications?**

Reviewer #1: See comments made on manuscript throughout.

Reviewer #2: see general comments

Reviewer #3: (No Response)

**Summary and General Comments**

Reviewer #1: I have examined the revised version of Yuan et al. regarding the effect of arsenuate (ART) on fibrosis originally induced by Schistosoma japonicum (see several comments made on the manuscript itself). The authors have responded to some of the suggestions made, but unfortunately, I find myself less convinced by this version than the original version regarding whether a true effect of ART has been detected. The paper remains frustratingly short of key details needed to fully evaluate the work. This is specified in many of the comments made throughout the paper, some being very similar to comments made on my first review. Some examples include the lack of provision of spectrophotometrically evaluation on the status of HSC cells in culture following ART treatment mentioned in the M&M. Instead a series of photographs of cells exposed to different concentrations of ART appears and its hard to know what kind of conclusion we are to draw from these photos – no other mention of spectrophotometric results is made. Graphs resulting from analyses of tissue sections to reveal, for example, changes in collagen production due to ART are provided but we are left without any idea for how the assessments were made and the associated figure legends do not provide the needed detail to indicate what some of the graphs are purported to show. Some of the bar graphs in figure 2C have extra columns in them that are not explained, and this might have helped sell their case. Another example was the issue of biopsies being done to alleviate the concern that PZQ might not have killed all the worms, a critical point since ongoing presence of live worms (and egg production) would have altered background levels of fibrosis. No details were provided for how the biopsies were obtained, or what effects that had on the subsequent sample sizes used in some of the histologically-based analyses. The transcriptomics results were based on n=1 or n=2, depending on the treatment group with each ART group represented by only one. As one who knows how frustrating it can be to get good replicates for transcriptomics studies, it would have been somewhat reassuring if it was at least clear that when there were two replicates obtained per treatment, they generally tended to be supportive of one another, or that the two ART treatments (one low and one high) trended the same way (a look at the heat maps suggested this was not strongly so). I was intrigued and impressed by the model put forth for how the authors’ results might add to our knowledge about how fibrosis might be influenced by ART, but the overall package is somewhat frustrating in its lack of strong treatment effects and continued lack of detail.

Reviewer #2: The critical concern in the present manuscript, where the central claim is the non-antiparasitic but purely anti-fibrotic potential of artesunate is critically hampered by a missing verification.

The PZQ treatment performed (75 mg/kg x 3 days) has not been unequivocally validated to completely clear the infection and stop the egg deposition. Light microscopy alone is not sufficient to make such a claim. In fact, the parenchyma of the liver tissues in figure 2C display more egg-like structures in the MOD group when compared to the ART groups that would argue for a reduced infestation of the livers in the ART groups. In fact, the conclusion of sterile treatment by PZQ, as used in this study, can only be drawn by perfusion to flush the eventual remnant adult worms and assess their viability and/or the tissue digestion with KOH to ensure that no further egg reduction takes place when PZQ-treated groups (MOD) are further treated with ART. In fact, such a reduction is known to be induced by ART (Journal of Helminthology , Volume 83 , Issue 1 , March 2009 , pp. 7 - 11) and would introduce biases to the present experimental design towards the revelation of the additional anti-parasitic effect of ART rather than a parasite-independent antifibrotic effect.

 As the data stands, it is equivocal that PZQ with doses as low as 75mg/kg administered orally three times would ensure full clearance of S japonicum worms thus interrupt egg deposition. The validation of this full clearance by PZQ, as used here, is critical to claim an anti-fibrotic and not just an anti-parasitic effect of ART. In fact, the use of an infection control (without PZQ treatment) would have shown to the authors that the anti-parasitic potential of PZQ can result in indirectly driving a reduction of the fibrotic process by removal of the etiological agent (Sci Rep. 2020 Jun 30;10(1):10638.; PLoS Negl Trop Dis. 2017 Feb; 11(2): e0005372.). 

Therefore, a validating assessment of the inability of the ART treatment, as used in this study, to further drive an anti-parasitic effect following a limited PZQ treatment, a well-known effect of ART on schistosome parasites (Trans R Soc Trop Med Hyg. 2002 May-Jun;96(3):318-23; Acta Trop. 2002 May;82(2):175-81.), is indispensable to support their claim of a true anti-fibrotic effect of ART independently from an antiparasitic effect.

This is the basis of all other analyses and conclusions made and as such is not dispensable

Reviewer #3: (No Response)

PLOS authors have the option to publish the peer review history of their article (what does this mean?). If published, this will include your full peer review and any attached files.

Reviewer #1: No

Reviewer #2: No

Reviewer #3: No
---

## [Editor Report · Decision Letter 2]

29 Aug 2023

Dear Dr. Kong,

We are pleased to inform you that your manuscript 'Mining Host Candidate Regulators of Schistosomiasis-Induced Liver Fibrosis in Response to Artesunate Therapy through Transcriptomics Approach' has been provisionally accepted for publication in PLOS Neglected Tropical Diseases.

Best regards,

Krystyna Cwiklinski, PhD

Academic Editor

Cinzia Cantacessi

Section Editor

The authors have addressed the comments raised by the reviewers, particularly in regards to expanding the discussion to include the relevant points. The manuscript is now suitable for publication.

---

## [Editor Report · Acceptance letter]

26 Sep 2023

Dear Dr. Kong,

We are delighted to inform you that your manuscript, "Mining Host Candidate Regulators of Schistosomiasis-Induced Liver Fibrosis in Response to Artesunate Therapy through Transcriptomics Approach," has been formally accepted for publication in PLOS Neglected Tropical Diseases.

Best regards,

Shaden Kamhawi

co-Editor-in-Chief

Paul Brindley

co-Editor-in-Chief
